# Use of Pranoprofen After Strabismus Surgery in Children

**DOI:** 10.3390/jcm14062104

**Published:** 2025-03-19

**Authors:** Wojciech Pawłowski, Beata Urban, Joanna Zawistowska, Alina Bakunowicz-Łazarczyk

**Affiliations:** Department of Paediatric Ophthalmology and Strabismus, Medical University of Bialystok, ul. Waszyngtona 17, 15-274 Bialystok, Poland; beata.urban@umb.edu.pl (B.U.); alina.bakunowicz-lazarczyk@umb.edu.pl (A.B.-Ł.)

**Keywords:** strabismus, strabismus surgery, complications, pranoprofen

## Abstract

**Objectives**: The aim of our study is to evaluate the efficacy of pranoprofen 0.1% in preventing the occurrence of postoperative complications, including postoperative ocular inflammation after strabismus surgery in children. **Methods:** 90 children operated on for strabismus in the Department of Paediatric Ophthalmology and Strabismus of the Medical University of Bialystok between 2022 and 2023 were included in the study. The patients were aged from 2 to 17 years old. Patients were divided into three groups of 30 patients according to the method of topical postoperative treatment (group I: tobramycin, pranoprofen, dexpanthenol; group II: tobramycin, dexamethasone, dexpanthenol; group III: tobramycin, pranoprofen, dexamethasone, dexpanthenol). Patients were followed up the day after surgery, on postoperative day 14 and then 3 months after surgery. **Results**: No child showed signs of postoperative infection. There was one case of allergic reaction each in groups I and III. Most complications were present in group II (six cases: there were allergic reactions that occurred in three patients. Two patients experienced a complication in the form of delle, while one patient developed a granuloma), which was a statistically significant difference compared to groups I and III (*p* = 0.032). **Conclusions**: Pranoprofen used postoperatively is as effective as a glucocorticosteroid, and from our observations, carries fewer complications in cases requiring prolonged use of the anti-inflammatory treatment. Systematic postoperative follow-up is essential for the diagnosis and possible management of postsurgical complications.

## 1. Introduction

Strabismus surgery, like many other ophthalmic procedures, is safe and effective, but significant complications may occur if appropriate intraoperative and postoperative care are not taken. According to the literature, the incidence of clinically significant postoperative infection is 1 in 1100 strabismus procedures [1]. Equally important is the interview conducted upon admission regarding the child’s general condition, chronic diseases, allergies, recent infections and medication taking. Before qualifying a patient for strabismus surgery, attention should be paid to appropriate preoperative preparation and possible treatment of local inflammations (e.g., blepharitis) as they may significantly affect wound healing. Both intraoperative and postoperative prophylaxis should be used to prevent unwanted inflammation and infections. By using topical antibiotics, it is assumed that the risk of infection will be diminished, while by using glucocorticosteroid formulations or non-steroidal anti-inflammatory drugs (NSAIDs) after strabismus surgery, it is assumed that postoperative inflammation will be minimized. Due to the general increase in antibiotic resistance and the fact that glucocorticosteroid preparations or NSAIDs after strabismus surgery are usually used slightly longer than antibiotics, it seems appropriate to prescribe separate formulations rather than combined drugs (antibiotic/glucocorticosteroid). This management will reduce the duration of antibiotic use and thus reduce the possibility of antibiotic resistance developing. The aim of our study is to evaluate the efficacy of pranoprofen 0.1% in combination with other ophthalmic medications in preventing the occurrence of postoperative complications, including ocular inflammation after strabismus surgery in children. We based our study on a group of 90 patients operated on for strabismus in the Department of Paediatric Ophthalmology and Strabismus of the Medical University of Bialystok between 2022 and 2023. Patients were divided into three groups of 30 patients according to the method of topical postoperative treatment (group I: tobramycin, pranoprofen, dexpanthenol; group II: tobramycin, dexamethasone, dexpanthenol; group III: tobramycin, pranoprofen, dexamethasone, dexpanthenol). Patients were followed up the day after surgery, on postoperative day 14 and then 3 months after surgery.

## 2. Materials and Methods

We conducted an observational study involving 90 patients operated on for strabismus at the Department of Paediatric Ophthalmology and Strabismus of the Medical University of Bialystok between 2022 and 2023. All participants were from Poland, comprising 43 boys and 47 girls. The patients were aged from 2 to 17 years old, with the majority falling within the 4 to 8 year age range. In the first group, there were 14 boys and 16 girls. In the second group, there were 15 boys and 15 girls. In the third group, there were 14 boys and 16 girls. The group of patients observed by us included patients undergoing surgery due to various types of strabismus. In the group of observed patients, there were 55 patients with convergent strabismus. 35 patients had divergent strabismus. Some patients had strabismus with an oblique component. After the initial interview at the time of admission to the procedure, we qualified children with no history of allergies to the group of patients we observed. During the interview, we made sure that the child had no signs of infection within the last 2 weeks. All children qualified for the procedure had no signs of eyelid margin infection. On the day of admission and before surgery, patients received antibiotic prophylaxis in the form of tobramycin eye drops. The surgical procedure was performed in accordance with aseptic principles. Careful preparation of the surgical site was a priority due to the possibility of complications in the form of inflammation associated with the patient’s microflora. In our case, preparation of the surgical site consisted of thoroughly washing the skin of the surgical site, the patient’s eyelashes and eyebrows with a 10% povidone–iodine solution and administering a 5% povidone–iodine solution into the conjunctival sac. Then, a sterile drape was used to cover the surgical site and the eyelashes were separated from the surgical site. The conjunctival sac was rinsed with a povidone–iodine solution and then Tobramycin drops were administered before the conjunctival incision. After the surgical procedure, Tobramycin ointment was administered into the conjunctival sac and a sterile dressing was applied. The average duration of strabismus surgery was 40 min and depended on whether the procedure was performed on a single muscle or multiple muscles. The day after surgery, the local condition was monitored using a slit lamp and the patients were discharged home with recommendations. Before the patients were discharged, the parents were appropriately instructed regarding the use of postoperative prophylaxis. Patients were divided into three groups of 30 patients according to the type of topical postoperative prophylaxis. The types of drops that were administered to each group of patients after strabismus surgery are shown in Table 1. The frequency of drug administration in each group was as follows: Tobramycin: 1 drop into the conjunctival sac of the operated eye 4 times daily for 14 days. Pranoprofen: 2 drops into the conjunctival sac of the operated eye 4 times daily for 14 days. Dexamethasone: 1 drop into the conjunctival sac of the operated eye 4 times daily for 14 days. Dexpanthenol: 1 drop in the conjunctival sac of the operated eye 4 times daily for 14 days.

Patients were followed up the day after surgery, on postoperative day 14 and then 3 months after surgery. The follow-up examination consisted of an assessment of visual acuity, bilateral and monocular double vision, eye movement and alignment. Above all, the anterior segment of both eyes was examined under a slit lamp, paying particular attention to the postoperative wound, its tightness, and the presence of abnormal discharge in the conjunctival sac. The fundus of both eyes was also examined.

We considered signs of infection and symptoms of postoperative complications as the following: delle, scleritis, erythema and swelling of the eyelids preventing opening of the eyelid stroma, presence of purulent discharge in the conjunctival sac, increased eye pain, presence of conjunctival abscess, Tenon’s pouch abscess, periorbital and orbital cellulitis and intraocular inflammation, stretched scar syndrome, fat adhesion syndrome or anterior ischaemia.

A chi-square test of independence was used to test for a relationship between the postoperative treatment used and the incidence of complications. The level of significance in this chapter was considered to be *α* = 0.05, meaning that results of *p* < 0.05 were treated as statistically significant.

Written informed consent for study participation was obtained from the participants or legal guardians of subjects under 16 years of age. The study was conducted in accordance with the Declaration of Helsinki Guidelines for Biomedical Research Involving Human Subjects. The study was approved by the local Ethics Committee of the Medical University of Bialystok, Poland (No. APK.002.415/2024).

## 3. Results

The relationship between the postoperative treatment used and the incidence of complications, regardless of their type, was tested. For this purpose, the Fisher–Freeman–Halton exact test was performed, as the assumptions for using Pearson’s chi-square test were violated (the number of expected values was less than 5 for 50% of the cells). Its results are presented in Table 2.

The result of the performed Fisher–Freeman–Halton test proved to be statistically significant (*p* = 0.046), indicating that there was a relationship between the type of postoperative treatment used and the incidence of complications. The strength of this relationship was weak, as indicated by a V-Kramer value below 0.30 (V(c) = 0.28). A pairwise comparison analysis was performed to assess the exact differences between the groups. This was done by analyzing the values of the adjusted standardized residuals. Values greater than |1.96| were obtained for the result in Group II, indicating significant differences between the complication rates in this group compared to the other two groups.

The specific type of complications occurring in each group was also analyzed. The outcome of the incidence analysis of postoperative complications is shown in Figure 1.

In patients who experienced complications, additional check-ups were recommended. In the case of the following complications, the duration of local treatment was extended. In the case of a delle-type complication, it was decided to extend the use of local dexamethasone and additionally apply pranoprofen and intensify moisturizing, achieving an improvement in the local condition after 10 days. The allergic reaction resolved after additional oral antihistamine administration. In the case of granuloma, improvement was also achieved with the addition of topical pranoprofen. All patients followed up 3 months after surgery did not demonstrate any local complications.

## 4. Discussion

Postoperative infection after ocular surgery can occur if sterility is breached or if the patient had a predisposition for developing inflammation, such as blepharitis or nasolacrimal duct stenosis. These conditions increase the number of bacteria at the surgical site. Most infections occur near the surgical incision of the conjunctiva and occur within the first week after surgery. Symptoms of infection include conjunctival-ocular injection, erythema and swelling of the eyelids, the presence of purulent discharge in the conjunctival sac and eye pain [1]. Infections following strabismus surgery include subconjunctival abscess, sub-Tenon’s abscess, periocular infection (periorbital and orbital cellulitis) and endophthalmitis [2,3]. The risk of postoperative infection is increased in very young patients, especially those with developmental delays, who may have difficulty using eye drops and with eyelid hygiene after surgery. Infection may also be associated with a history of skin or middle ear infection and acute or chronic rhinosinusitis. The risk of postoperative infection can be minimized by treating any infection prior to surgery.

Pranoprofen is a tricyclic NSAID, a propionic acid derivative. Pranoprofen can inhibit the inflammatory response mainly by inhibiting cyclooxygenase 1 and 2 activity, thereby blocking the formation of eicosatetraenoic acid derivatives and inhibiting prostaglandin synthesis [4,5]. In addition, it inhibits thrombopoietin, tumor necrosis factor (TNF), protein kinase, histamine and bradykinin, making its indications broad. After topical administration into the conjunctival sac, it shows the highest concentration in the conjunctiva and cornea after 30 min [6].

As a NSAID, pranoprofen reduces inflammatory reactions but also prolongs the tear film break up time, thereby alleviating symptoms associated with dry eye [6]. Numerous studies have shown that pranoprofen, in combination with sodium hyaluronate, tobramycin and dexamethasone, acts synergistically in reducing the inflammatory response of the eye and shortening the healing time at the incision site, thus maintaining adequate moisture in the eye, promoting healing of the conjunctiva and cornea and improving circulation in the eye to achieve an optimal healing effect [7]. Pranoprofen is used in conditions such as blepharitis, conjunctivitis, keratitis and scleritis [4]. Furthermore, experimental studies have shown that the application of pranoprofen in the early stages of principle burns leads to a reduction in interleukin-1β (IL-1β) production by inhibiting the expression of Nod-like receptor protein 3 (NLRP3) inflammasomes [4]. In addition, pranoprofen can inhibit the expression of matrix metallopeptidase 13 (MMP-13), which is closely associated with the process of neovascularization after burn injuries [4]. Pranoprofen has also found use in the treatment of chronic allergic conjunctivitis, acute central serous chorioretinopathy, age-related macular degeneration, cystoid macular oedema, pterygium and dry eye [7,8,9,10]. Due to the inhibition of prostaglandin release by pranoprofen, it has found use in relieving the inflammatory reaction in the anterior segment of the eye and postoperative pain, reducing tissue congestion and swelling after cataract surgery [11]. As it turns out, the preoperative administration of pranoprofen eye drops reduced the perceived pain during second-eye cataract surgery. Monocyte chemoattractant protein 1 (MCP-1), a pain-related cytokine, was associated with the pain-relieving mechanism of pranoprofen when second-eye surgery was performed 1 week after second-eye surgery [11].

Many ocular surface and anterior segment conditions have an inflammatory component, so topical NSAIDs may be a therapeutic option for these conditions. These drugs have been successfully used after ophthalmic surgery due to their anti-inflammatory and analgesic properties, including after strabismus surgery [12]. The therapeutic effect of pranoprofen was used in our study, where tobramycin, dexamethasone and dexpanthenol were administered in group III patients in addition to pranoprofen; the division of patients into groups according to the drops used is shown in Table 1.

Analysis of the incidence of postoperative complications for each group shows that they occurred significantly more frequently in patients in group II (20%) (*p* = 0.046), i.e., in 6 of the 30 patients treated with tobramycin, dexamethasone and dexpanthenol, than in the other two groups (both 3.3%) (Figure 1). It is worth emphasizing that in the pediatric population, each complication that requires additional treatment is difficult to work with. Therefore, it is worth striving to achieve the best possible treatment effect using the smallest possible number of preparations. Therefore, adapting the appropriate postoperative prophylaxis has a huge impact on the local condition and also the general condition of the patient.

A clinically significant (requiring hospitalization) postoperative infection was found in one child at the first follow-up visit, 14 days after strabismus surgery. This was due to a lack of adequate compliance with the post-operative instructions, which was admitted by the parents of the operated child. During the follow-up, it was found that the child had had the operated eye taped for 2 weeks and had not been given any drops, and was therefore excluded from further study participation. Fortunately, infectious complications after strabismus surgery are relatively rare. In a study by Heo et al., this complication was found in 167 patients among 151011 strabismus surgeries in U.S. claims databases, representing 0.111% of strabismus surgeries [2]. Interestingly, a recent large database study has shown that the prescription of prophylactic topical antibiotics did not decrease rates of surgical site infections following strabismus surgery [13].

Another common complication following strabismus surgery is an allergic reaction. It is usually caused by a reaction to sutures (e.g., polyglactin sutures) [14]. In our study, an allergic reaction occurred in five children, accounting for 62.5% of all complications. An allergic reaction occurred in one patient in group I, in three patients in group II and in one patient in group III. At the same time, in group II, where the most complications occurred, the most common was an allergic reaction, which occurred in three patients. This complication accounted for half of all complications in group II, which was also 10% of the total in this group. The allergic reaction was resolved with the addition of an antihistamine. It is worth noting that pranoprofen is just as effective as fluorometholone for the management of chronic allergic conjunctivitis [8].

Another complication was delle observed in two patients in group II, representing 6.7% of the total group and 33.3% of all complications. Improvement in the local condition was achieved with intensive use of moisturizers and the addition of topical pranoprofen. Another observed complication after strabismus surgery was granuloma, which occurred in one patient in group II, representing 3.3% of the total group and 16.7% of all complications. Improvement was also achieved with the addition of topical pranoprofen.

In the study by Bradbury and Tayler, the most common reported complications after strabismus surgery were as follows: perforation of the globe (0.08%), followed by a suspected slipped muscle (0.067%), severe infection (0.06%), scleritis (0.02%) and lost muscle 0.02%) [15]. In our study, we did not observe any of these complications. We also did not find complications such as stretched scar syndrome, fat adherence syndrome or anterior segment ischemia, which are described after strabismus surgery [3]. This may be due to the relatively small number of patients included in the study (90 patients) and the relatively short follow-up time (3 months).

Postoperative prophylaxis usually involves topical antibiotic therapy and steroid drops or drops containing NSAIDs. In addition, gel preparations containing dexpanthenol are helpful for wound healing. Glucocorticosteroids, which are the first-line drugs in ophthalmology for the treatment of inflammation, may have many side effects, e.g., increased intraocular pressure, inhibition of corneal epithelial repair and exacerbation of infection. In an analysis by Hinokuma, in patients with postoperative cataract surgery, pterygium and patients with epithelial inflammation, the use of topical steroids resulted in healing corneal erosions and increased intraocular pressure. The resolution of side effects and resolution of inflammation was only achieved after steroid discontinuation and the inclusion of topical pranoprofen [16]. Other studies have shown that pranoprofen significantly improves corneal epithelial cell migration and accelerates healing compared to diclofenac and fluorometholone [17]. It is worth mentioning the study by Sawa et al. who demonstrated that pranoprofen may be as effective in treating ocular surface disorders as bromophenac, with a lower incidence of adverse effects in the pranoprofen-treated group than in the bromophenac-treated patients [18].

A 2019 survey of strabismus surgery specialists by the American Association for Pediatric Ophthalmology and Strabismus (AAPOS) found that 90% of them administered topical antibiotics (both with and without steroids) at the end of the procedure, and 85.5% recommended antibiotic therapy postoperatively [19]. Such conduct demonstrates awareness of the occurrence of postoperative infections that are sometimes difficult to control, which is why most surgeons who want to avoid this type of complication continue to use and will continue to use antibiotics even when appropriate perioperative asepsis is used.

## 5. Conclusions

Pranoprofen used postoperatively is as effective as a glucocorticosteroid, and from our observations, carries fewer complications in cases requiring prolonged use of the anti-inflammatory drug. Due to increasing antibiotic resistance, it is worth considering separate preparations containing an antibiotic, a glucocorticosteroid or a non-steroidal anti-inflammatory drug. Systematic postoperative follow-ups are essential for the diagnosis and possible treatment of postoperative complications.

## Figures and Tables

**Figure 1 jcm-14-02104-f001:**
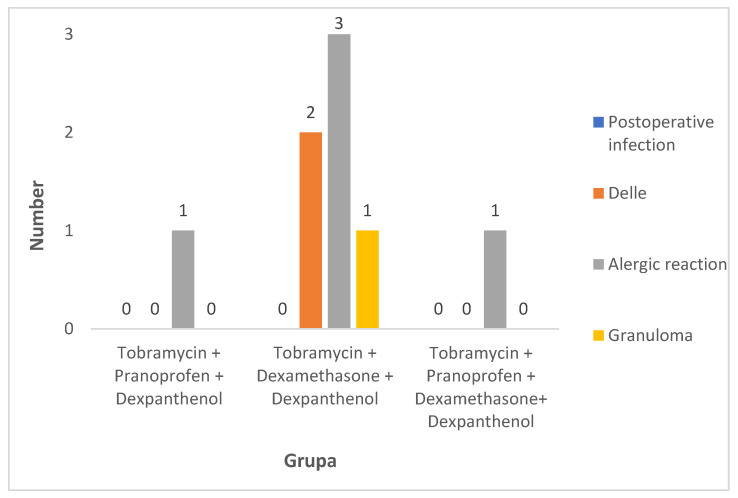
Incidence of specific complications in the groups analyzed. Group I: tobramycin, pranoprofen, dexpanthenol; Group II: tobramycin, dexamethasone, dexpanthenol; Group III: tobramycin, pranoprofen, dexamethasone, dexpanthenol.

**Table 1 jcm-14-02104-t001:** Topical treatment administered in each patient group.

Group	I (30 Patients)	II (30 Patients)	III (30 Patients)
Treatment	Tobramycin Pranoprofen Dexpanthenol	Tobramycin Dexamethasone Dexpanthenol	Tobramycin Pranoprofen Dexamethasone Dexpanthenol

**Table 2 jcm-14-02104-t002:** Relationship between the type of postoperative treatment administered and the incidence of complications—Fisher–Freeman–Halton exact test.

Postoperative Complications		Group I	Group II	Group III	χ^2^	*p*	*V* _c_
No	**N**	29	24	29	5.60	**0.046**	0.28
	**%**	96.70%	80.00%	96.70%			
	Residual ^a^	1.3	−2.6	1.3			
Yes	**N**	1	6	1			
	**%**	3.30%	20.00%	3.30%			
	Residual ^a^	−1.3	2.6	−1.3			
Total	**N**	30	30	30			
	**%**	100%	100%	100%			

N—number of observations; χ^2^—chi-square test result; *p*—statistical significance; *V*(c)—V-Kramer effect strength index. Group I: tobramycin, pranoprofen, dexpanthenol; Group II: tobramycin, dexamethasone, dexpanthenol; Group III: tobramycin, pranoprofen, dexamethasone, dexpanthenol. ^a^ Adjusted standardized residuals.

## Data Availability

The original contributions presented in this study are included in the article. Further inquiries can be directed to the corresponding author(s).

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
