# Peer review of "Use of Pranoprofen After Strabismus Surgery in Children"

_jcm, 2025, doi:10.3390/jcm14062104_

Round 1
Reviewer 1 Report
Comments and Suggestions for Authors
This study compared pranoprofen 0.1% and dexamethasone for postoperative complications in 90 children after strabismus surgery. Group I (pranoprofen) and III (pranoprofen + dexamethasone) had fewer complications (1 case each) vs. Group II (dexamethasone, 6 cases; p=0.032), with no infections reported. Pranoprofen showed anti-inflammatory efficacy comparable to steroids but with fewer complications in prolonged use. However, the following aspects could be improved to further enhance the manuscript:
- The method of randomization and blinding (if any) is not described. Potential selection bias may affect the validity of the statistically significant difference between groups.Provide the mean/median age and distribution (e.g., quartiles) for each group.
- To enhance the rigor and interpretability of your findings, we kindly encourage the authors to supplement a baseline characteristics table detailing the distribution of age, sex, preoperative inflammation severity (e.g., grading scales), and surgical details (e.g., number of muscles operated, operative duration) across all three groups. This will help confirm the comparability of groups and assess potential confounding.
- While a chi-square test was used to compare complication rates across groups, critical limitations exist:The chi-square test assumes expected cell frequencies ≥5 in contingency tables. Given the low complication counts (1 case in Groups I/III vs. 6 in Group II), the test’s validity is questionable. A Fisher’s exact test or alternative methods (e.g., Monte Carlo simulation) would be more appropriate for sparse data.
- The chi-square test only evaluates global differences between groups. To identify which specific pairs of groups differ (Group II vs. I/III), post-hoc pairwise comparisons (with corrections for multiple testing, e.g., Bonferroni) are necessary. Without this, the reported p-value (p=0.032) overstates the evidence for between-group differences.
- The reported Cramer’s V value of 0.28 indicates a weak association between treatment type and complications. However:Statistical significance (p=0.032) does not imply clinical relevance. The clinical importance of reducing complications from 20% (6/30 in Group II) to 3.3% (1/30 in Groups I/III) should be explicitly discussed.
- Reporting absolute risk reduction (ARR) or relative risk (RR) with confidence intervals would better contextualize the clinical significance of the findings.
- The sentence "After topical administration into the conjunctival sac, it shows the highest concentration in the conjunctiva and cornea after 30 minutes" requires citation to substantiate the pharmacokinetic claims.
- The sentence "An allergic reaction occurred in 1 patient in group I, in 3 patients in group II and in one patient in group III." uses both Arabic numerals (1) and written-out words (one). Please ensure consistent formatting.
Author Response
Question 1 : The method of randomization and blinding (if any) is not described. Potential selection bias may affect the validity of the statistically significant difference between groups.Provide the mean/median age and distribution (e.g., quartiles) for each group.
Answer: Dear Reviewer,
Thank you for your valuable feedback. We would like to clarify that our study is an observational analysis conducted post-surgery, focusing on all children aged 2 to 17 years. Given the observational nature of our research, randomization and blinding were not applicable. To address potential selection bias, we have provided demographic information, including the mean and median ages to the revised manuscript. We appreciate your insights and believe that this addition information enhances the transparency and validity of our findings.
Question 2: To enhance the rigor and interpretability of your findings, we kindly encourage the authors to supplement a baseline characteristics table detailing the distribution of age, sex, preoperative inflammation severity (e.g., grading scales), and surgical details (e.g., number of muscles operated, operative duration) across all three groups. This will help confirm the comparability of groups and assess potential confounding.
Answer: Thank you for your suggestion. We acknowledge the importance of providing a comprehensive baseline characteristics to assess the comparability of groups and identify potential confounding factors. In response, we have supplemented the revised manuscript with information detailing the distribution of age, sex, operative duration time across all three groups. We believe this addition strengthens the validity of our study.
Question 3: While a chi-square test was used to compare complication rates across groups, critical limitations exist:The chi-square test assumes expected cell frequencies ≥5 in contingency tables. Given the low complication counts (1 case in Groups I/III vs. 6 in Group II), the test’s validity is questionable. A Fisher’s exact test or alternative methods (e.g., Monte Carlo simulation) would be more appropriate for sparse data.
Answer: Thank you for the insightful observation. Therefore, we have improved the analysis using the Fisher-Freeman-Halton exact test, which is used when the expected cell counts are missing. The p-value for this test is p = 0.046.
Question 4: The chi-square test only evaluates global differences between groups. To identify which specific pairs of groups differ (Group II vs. I/III), post-hoc pairwise comparisons (with corrections for multiple testing, e.g., Bonferroni) are necessary. Without this, the reported p-value (p=0.032) overstates the evidence for between-group differences.
Answer: The suggestion about the need for a pairwise comparison test is correct. Accordingly, we added the results with such an analysis. However, we chose to make these comparisons based on the values of the adjusted standardized residuals. The use of a statistical test (the Bonferroni corrected Z-test) in this situation is not a good option in our opinion, since the power of those tests would be much too low.
Question 5: The reported Cramer’s V value of 0.28 indicates a weak association between treatment type and complications. However:Statistical significance (p=0.032) does not imply clinical relevance. The clinical importance of reducing complications from 20% (6/30 in Group II) to 3.3% (1/30 in Groups I/III) should be explicitly discussed.
Answer: Thank you for your comment. We agree that although the strength of the tested effect was weak, the decrease in the percentage of complications from 20% to 3.3% can be considered significant, because this difference may have great practical and clinical significance. Therefore, in the discussion we emphasized the importance of the occurrence of these complications and the difficulties in their treatment, especially in the pediatric group studied by us.
Question 6: Reporting absolute risk reduction (ARR) or relative risk (RR) with confidence intervals would better contextualize the clinical significance of the findings.
Answer: In our study, we conducted a chi-square test of independence, and based on the analyses performed, there is little more that can be done within this framework. To follow your suggested direction, a different research model would be required, which would necessitate a new study. Given the nature of our data, we do not have the necessary structure to implement this approach. Therefore, we are unable to propose any additional analyses as a supplement to the current study. We appreciate your insights and welcome any further suggestions.
Question 7: The sentence "After topical administration into the conjunctival sac, it shows the highest concentration in the conjunctiva and cornea after 30 minutes" requires citation to substantiate the pharmacokinetic claims.
Answer: Thank you very much for your insightful observation. The manuscript has been updated with the appropriate citation.
Question 8: The sentence "An allergic reaction occurred in 1 patient in group I, in 3 patients in group II and in one patient in group III." uses both Arabic numerals (1) and written-out words (one). Please ensure consistent formatting
Answer: Dear Reviewer, thank you very much for your insightful comment. The sentence has been revised and incorporated into the revised manuscript.
Reviewer 2 Report
Comments and Suggestions for Authors
This study offers a valuable investigation into the efficacy of pranoprofen 0.1% in preventing postoperative complications following strabismus surgery in children, a significant concern in pediatric ophthalmology. The authors aim to assess the impact of pranoprofen on postoperative ocular inflammation and other complications, providing a direct comparison with other treatment protocols. The methodology is robust, involving a well-defined cohort of 90 children, aged 2 to 17, who were divided into three treatment groups. The study’s design is clear and the follow-up schedule—one day, 14 days, and three months after surgery—ensures thorough assessment of postoperative outcomes.
The results are concisely presented, with no signs of postoperative infection in any of the children. They indicate the potential advantage of pranoprofen in minimizing complications compared to dexamethasone, particularly in cases requiring extended anti-inflammatory treatment. While the conclusions are clear, suggesting that pranoprofen is as effective as glucocorticosteroids with fewer complications, the discussion could benefit from a deeper exploration of the mechanisms behind the observed differences between treatment groups. Additionally, the study might consider addressing the limitations of the study design, such as the lack of blinding or randomization, which could impact the generalizability of the findings.
Overall, the study provides important insights into postoperative care for pediatric strabismus surgery. The authors present strong evidence that pranoprofen could be a safer alternative to glucocorticosteroids, offering a promising avenue for reducing complications. A more detailed exploration of treatment mechanisms and further validation through randomized controlled trials would strengthen the study’s conclusions.
Author Response
Dear Reviewer,
Thank you for your through review and valuable feedback. We appreciate your insightful comments of our study and its implications for postoperative care in pediatric strabismus surgery. Below, we address your key points:
- Exploration of Mechanisms behind treatment differences: In response we have expanded the discussion section to include a more detailed analysis of possible pharmacodynamic and anti-inflammatory effects of pranoprofen in comparison to gluococoricosteroids. This addition provides a cleared perspective on how these treatments may contribute to varying postoperative outcomes. We agree that although the strength of the tested effect was weak, the decrease in the percentage of complications from 20% to 3.3% can be considered significant, because this difference may have great practical and clinical significance. Therefore, in the discussion we emphasized the importance of the occurrence of these complications and the difficulties in their treatment, especially in the pediatric group studied by us.
- Study design and limitations: As you correctly pointed out, our study is observational and does not include blinding or randomization. We have explicitly stated that in the discussion section and emphasized that our findings should be interpreted within this context. While randomization controlled trials (RCTs) would provide stronger validation, our study offers valuable preliminary evidence supporting the safety and efficacy of pranoprofen in postoperative care.
- Generalizability of Findings: We recognize the potential limitations in generalizing our results to broader populations. To address this, we have now included a more detailed breakdown of the demographic and clinical characteristics of our study participants in the baseline characteristics. This addition ensures greater transparency regarding the composition of the study population and helps assess the applicability of our findings.
We appreciate your recognition of the potential advantages of pranoprofen. While our study suggests it could be a safer alternative to glucocorticosteroids, we acknowledge that further clinical trials are necessary to confirm its superiority in terms of long-term safety and efficacy.
We sincerely appreciate your constructive feedback, which has allowed us to refine our manuscript. We believe that the revisions enhance the clarity, rigor, and scientific value of our study.
Round 2
Reviewer 1 Report
Comments and Suggestions for Authors
The authors addressed my concerns.
Author Response
Thank you very much for accepting the manuscript after incorporating your comments. I truly appreciate your time and consideration.